# The MEK/ERK Module Is Reprogrammed in Remodeling Adult Cardiomyocytes

**DOI:** 10.3390/ijms21176348

**Published:** 2020-09-01

**Authors:** Kubin Thomas, Cetinkaya Ayse, Kubin Natalia, Bramlage Peter, Sen-Hild Bedriye, Gajawada Praveen, Akintürk Hakan, Schönburg Markus, Schaper Wolfgang, Choi Yeong-Hoon, Barancik Miroslav, Richter Manfred

**Affiliations:** 1Department of Cardiac Surgery, Kerckhoff Heart Center, Benekestrasse 2-8, 61231 Bad Nauheim, Germany; a.cetinkaya@kerckhoff-klinik.de (C.A.); nataliajudo@aol.de (K.N.); gajawada@outlook.com (G.P.); m.schoenburg@kerckhoff-klinik.de (S.M.); y.choi@kerckhoff-klinik.de (C.Y.-H.); 2Campus Kerckhoff, Justus-Liebig-University Giessen, 61231 Bad Nauheim, Germany; 3Institute for Pharmacology and Preventive Medicine, Bahnhofstraße 20, 49661 Cloppenburg, Germany; peter.bramlage@ippmed.de; 4Pediatric Heart Center, Justus Liebig University, Feulgenstrasse 10-12, 35392 Giessen, Germany; bedriye.sen-hild@chiru.med.uni-giessen.de (S.-H.B.); hakan.akintuerk@chiru.med.uni-giessen.de (A.H.); 5Max-Planck-Institute for Heart and Lung Research, 61231 Bad Nauheim, Germany; wolfgang.schaper@mpi-bn.mpg.de; 6German Center for Cardiovascular Research (DZHK), Partner Site RhineMain, 60590 Frankfurt/Main, Germany; 7Centre of Experimental Medicine, Institute for Heart Research, Slovak Academy of Sciences, 84104 Bratislava, Slovakia

**Keywords:** Ras, Rap, Raf, MEK, ERK, remodeling, reprogramming, dedifferentiation, hypertrophy, heart failure

## Abstract

Fetal and hypertrophic remodeling are hallmarks of cardiac restructuring leading chronically to heart failure. Since the Ras/Raf/MEK/ERK cascade (MAPK) is involved in the development of heart failure, we hypothesized, first, that fetal remodeling is different from hypertrophy and, second, that remodeling of the MAPK occurs. To test our hypothesis, we analyzed models of cultured adult rat cardiomyocytes as well as investigated myocytes in the failing human myocardium by western blot and confocal microscopy. Fetal remodeling was induced through endothelial morphogens and monitored by the reexpression of Acta2, Actn1, and Actb. Serum-induced hypertrophy was determined by increased surface size and protein content of cardiomyocytes. Serum and morphogens caused reprogramming of Ras/Raf/MEK/ERK. In both models H-Ras, N-Ras, Rap2, B- and C-Raf, MEK1/2 as well as ERK1/2 increased while K-Ras was downregulated. Atrophy, MAPK-dependent ischemic resistance, loss of A-Raf, and reexpression of Rap1 and Erk3 highlighted fetal remodeling, while A-Raf accumulation marked hypertrophy. The knock-down of B-Raf by siRNA reduced MAPK activation and fetal reprogramming. In conclusion, we demonstrate that fetal and hypertrophic remodeling are independent processes and involve reprogramming of the MAPK.

## 1. Introduction

Cardiomyopathies, disorders of the heart with a broad spectrum of etiologies, develop through chronic cardiac remodeling. The term “cardiac remodeling” describes “changes that result in rearrangement of normally existing structures” [1]. “Cardiac remodeling” is the morphological change through a period of phenotypical adaptation to increased chronic loads, ultimately leading to heart failure. Cardiac remodeling is also an important response to the limited capacity of adult cardiomyocytes to proliferate and to replace lost myocytes in the damaged myocardium.

Since heart failure can no longer be considered a simple contractile disorder, the need for drugs addressing three significant therapeutic challenges is apparent: First, drugs that support beneficial remodeling to accelerate healing after acute cardiac damage such as myocardial infarction; second, drugs that delay or prevent deleterious remodeling such as fibrosis or inflammatory processes which are not yet out of control; and third, drugs that reverse an already present status of severe cardiac damage. The first two challenges might be therapeutically addressed through the application of inhibitor/activator compounds targeting key signaling cascades during cardiac adaptation and remodeling. At the same time, lost or damaged tissue might be replaced by tissue or cell transplantation.

Cardiac adaptation and remodeling are controlled by complex networks of signaling pathways which appear to be evolutionarily conserved in its multiple points of bifurcation, crosstalk, and feedback [2,3,4]. The reorganization of transduction cascades might lead to different signal qualities and might be an essential component in the regenerative capacity of cardiac patients. Thus, drug treatment at various stages of cardiac remodeling will automatically have a different impact on the therapeutic outcome by being either beneficial or detrimental to cardiac cells due to their ability to modulate the signaling input. Regarding the potential reorganization of signal transduction pathways, heart failure may even develop in drug-treated people with unrelated cardiac diseases such as cancer patients, raising the question of whether a chronic reprogramming of control points in signaling multiprotein complexes exists. If so, incoming biochemical clues will result in different signaling qualities and their understanding might help to appreciate the cardiotoxic effects of anti-cancer drugs [5,6,7].

The understanding of pathological regulation of the Ras/Raf/MEK/ERK cascade in muscle cells is of particular importance as it was shown that this pathway is involved in forelimb muscle development, arteriogenesis, heart failure, and myocardial infarction [8,9,10,11,12,13,14,15,16,17]. Importantly, the Ras/Raf/MEK/ERK cascade does not necessarily act in a linear way, as for example, B-Raf mediates muscle precursor cell migration independently from ERK activation [15]. As known from cancer therapy, deregulated cells might also develop resistance to certain Raf/MEK inhibitors, indicating a reduced susceptibility due to reprogramming of this signaling cascade [18,19]. Furthermore, modulation of activation or inhibition sites in the master gatekeepers MEK1/2 might not necessarily follow predicted phosphorylation patterns, indicating a much more complex regulation of this well-studied pathway [17,20,21].

We hypothesized that reorganization of signaling mediators downstream of growth factor/cytokine/chemokine receptors leads to different signaling qualities and might be an essential component in the development or prevention of heart failure. Therefore, we asked whether reprogramming of the Ras/Raf/MEK/ERK pathway as one of the major signaling cascades in cardiomyocytes is possible under conditions of chronic structural remodeling through culture-induced hypertrophic or fetal remodeling. Here, we define hypertrophic remodeling as a reorganization of cardiomyocyte structure and protein accumulation with limited or no reexpression of fetal genes. During fetal remodeling, the opposite occurs. The quality of remodeling was monitored by structural proteins known to be upregulated as well as reexpressed during the development of heart failure in human patients as well as in animal models. Remodeling of signaling cascades in adult cardiomyocytes has to the best of our knowledge not yet been described. For simplicity, we will refer here to the Ras/Raf/MEK/ERK cascade as MAPK.

## 2. Results

### 2.1. Reexpression of Fetal Genes, Cellular Morphology, and Atrophy Distinguish Fetal-Reprogrammed from Hypertrophic Cardiomyocytes

We first analyzed vinculin, α-tubulin, and desmin in cultured cardiomyocytes since these are increased in the failing human heart [22,23]. Serum, as well as morphogen treatment, led to a significant upregulation of α-tubulin, desmin, and vinculin after 10 and 20 days in cultured cardiomyocytes (Figure 1A,B). In general, morphogens exerted stronger effects than serum, indicating a higher remodeling capacity. Fetal remodeling was monitored by the reexpression of α-smooth muscle actin (SM-actin, Acta2), α-actinin-1 (Actn1), and β-actin (Actb). All three proteins are usually not expressed in normal adult cardiomyocytes (Figure 1A,B). Reexpression of α-SM-actin has been reported in a variety of animal models and patients with aortic stenosis [24]. SM-actin accumulation was detectable mainly in fetal remodeling myocytes (Figure 1A,B). Morphogens also induced the reexpression of α-actinin-1 and β-actin, but both proteins were barely detectable in serum-treated groups (Figure 1A,B). α-Actinin-1 appears to be a new marker of cardiac remodeling since it is reexpressed in cardiomyocytes of patients with dilated cardiomyopathy, aortic stenosis, and myocardial infarction, as well as in mice with myocarditis, dilated cardiomyopathy, and infarction [11,12,13,16,25,26,27]. Skeletal actin is present in adult rat cardiomyocytes and was downregulated by serum at day 10 but recovered thereafter.

Figure 2 demonstrates that morphogens and serum induce quite dramatic morphological changes during a 6 d and 20 d culture period. Freshly isolated cardiomyocytes show the typical rod-shaped morphology with a pronounced 3-dimensional structure and a prominent sarcomeric cross-striation already visible by light microscopy (Figure 2C). Upon serum stimulation, cardiomyocytes round up, increase their length and diameter (Figure 2A and Figure 3A,B), and form a tissue-like contracting monolayer until day 20 (Figure 2C). In contrast, morphogen-treated cardiomyocytes reestablish cell–cell contact by the formation of extensions until day 6 (Figure 2A and Figure 3A,B) and appear degenerated at day 20 (Figure 2C). In contrast to serum, the number of SM-actin positive cells increased to almost 40% of the cell population and was significantly downregulated by the MAPK inhibitor UO126 (Figure 2A,B). Morphogen-treated cultures showed massive cell lengthening but the diameter increased only slightly and total protein content decreased, indicating atrophy (Figure 3A–C). The difference in protein accumulation between the 0 d reference, serum, and morphogen can be explained by a pronounced cardiomyocyte flattening after morphogen treatment [28].

Inhibition of the MAPK reduced morphogen-induced cell lengthening but did not change the protein content. Knock-down of B-Raf by siRNA significantly reduced dedifferentiation and cell lengthening but had no effect on the protein content and cell diameter, indicating that MAPK activation plays a major role in morphogen induced fetal remodeling (Figure 2A,B and Figure 3A,B). In contrast, inhibition of the MAPK had no impact on serum-induced cell lengthening, increases in diameter, and protein accumulation (Figure 3A–C). Surprisingly, apart from inducing some contractile activities, we did not observe any morphological effect of isoproterenol, indicating that costimulants such as serum are essential for hypertrophic activities of this β-adrenoreceptor agonist (Figure 3A,B).

### 2.2. Morphogens Induce MAPK Dependent Ischemic Resistance and Fetal Remodeling

The loss of the original cardiomyocyte morphology and dedifferentiation is an adaptation to environmental challenges. As we previously demonstrated, the Ras/Raf/MEK/ERK pathway exerts dedifferentiation [11], and we questioned whether this cascade mediates cardiomyocyte protection under ischemic conditions (Figure 3D). Serum induced a 26% cell survival and was not substantially changed after addition of the MEK1/2 inhibitor 5 µM UO126. Morphogen treatment increased the number of surviving cells up to 46%, which dropped in the presence of inhibitor to 31%. Knock-down of B-Raf reduced the amount of surviving cardiomyocytes from 39% to 21% in the morphogen-treated groups. Note that these groups were pretreated with siCon and siB-Raf for three days before morphogen stimulation.

Morphogens, but not serum, induced ERK1/2 phosphorylation after 10 min and SM-actin reexpression after 6 days. MAPK activation and SM-actin reexpression were strongly reduced by the MEK1/2 inhibitor UO126 (Figure 4A). The MAPK inhibitor had little impact on the morphology of serum-stimulated cells but downregulated strongly dedifferentiation of morphogen-treated cardiomyocytes visible in the reduction of the number of SM-actin positive cells as well as in the total amount of SM-actin per culture (Figure 2A,B and Figure 4A). A certain degeneration of cardiomyocytes is visible in unstimulated control cells during the six-day culture period because unloading of cardiomyocytes led to a significant reduction in protein content and ERK1/2 phosphorylation with time (Con 0 d vs. Con 6 d; Figure 3C and Figure 4B).

### 2.3. B-Raf Is Responsible for MAPK Mediated Fetal Remodeling and Ischemic Resistance

Under the Raf isoforms B-Raf is considered as the major MAPK activator due to its strong biochemical activity [29,30]. In order to address specifically the function of this RAF isoform we utilized siRNA knock-down. Before stimulation with morphogens cultures were pretreated with siRNA for three days. B-Raf knock-down reduced significantly the number of α-smooth muscle actin (SM-actin)-positive cells (Figure 2A,B), cell lengthening (Figure 3A,B), and the total amount of SM-actin per culture (Figure 4C) until day six. Similarly, expression of the non-muscle α-actinin-1 (actinin-1) decreased sharply (Figure 4C). However, knock-down of B-Raf had no effect on the total protein content (Figure 3C).

In order to determine the influence of B-Raf on MEK/ERK activation, 6-day old cultures were stood for an additional day in basic medium and then stimulated with morphogens for 10 min. Knock-down of B-Raf reduced significantly the amount of phosphorylated MEK1/2 (P-MEK1/2) and ERK1/2 (P-ERK1/2), indicating that this Raf isoform is a major activator of the MAPK pathway in dedifferentiating adult cardiomyocytes (Figure 4C). In order to determine whether the sharp increase in the amount of B-Raf after morphogen treatment goes along with chronic remodeling of the MAPK cascade, we determined single components of this pathway by western blot analysis.

### 2.4. N-Ras Replaces K-Ras while H-Ras Increases Slightly during Remodeling

The classical K-Ras, H-, and N-Ras are regarded as entry points to the MAPK cascade. K-Ras is ubiquitous expressed, but H-Ras and N-Ras are also widely distributed. We detected all three Ras subfamily members in normal adult cardiomyocytes (Con 0 d; Figure 5A,B). Serum and morphogen treatment showed comparable effects on the pattern of Ras protein expression, but effects were more pronounced after morphogen treatment (Figure 5). H-Ras increased approximately 2- to 3-fold until day 10 and stayed relatively constant in both models over a 20-day period. N-Ras was elevated 4- and 6-fold in serum stimulated cardiomyocytes while it increased 10- and 11-fold in morphogen treated cultures after 10 and 20 days, respectively. In contrast, K-Ras protein was downregulated to 25% by serum and to 6 % in dedifferentiating cardiomyocytes during a 20-d culture period.

### 2.5. Rap1 Is Reexpressed during Fetal Remodeling

Rap1 and Rap2 belong to the same subfamily of Ras-related kinases. It has been shown that Rap1, like Ras, can activate the MAPK cascade and shares extensive homology within the regions involved in effector binding with Ras [31]. Here we demonstrate that Rap1 is massively reexpressed in morphogen-stimulated cardiomyocytes and almost 6-fold stronger than in hypertrophic cultures, but undetectable in 0 d reference cardiomyocytes (Figure 5). In contrast, Rap2 is clearly present in reference control cells with an approximately 3-fold increase in both remodeling models (Figure 5).

### 2.6. B-Raf and C-Raf Increase in Both Model Systems while A-Raf Increases Sharply During Hypertrophic Remodeling

Raf isozymes are considered as the main effectors of GTP-bound Ras signaling [32]. However, the functions of each Raf in adult cardiomyocytes are unclear with poor knowledge of the role of B-Raf during cardiac remodeling. C-Raf increases slightly in both models, which was more pronounced in morphogen-treated cultures (Figure 5). B-Raf was barely detectable in controls (reference 0 d; Figure 5). While the 95 kDa B-Raf was markedly elevated with time in both remodeling systems, its level was almost two times higher in fetal remodeling cardiomyocytes than in hypertrophic cells at day 10. Surprisingly, A-Raf was downregulated to 0.1-fold in fetal remodeling myocytes, but increased significantly approximately 3-fold in serum-treated cultures in a 20 day culture period (Figure 5). Despite strong elevations of A-Raf in all hypertrophic cultures, differences between peaks of A-Raf expression were observed. Some peaks were detectable at day 10, and others were observed at day 20.

### 2.7. MEK1/2 Increased Slightly and ERK1/2 Markedly in Both Models, but ERK3 Was Only Reexpressed during Fetal Remodeling

Raf activation leads to phosphorylation of MEK1/2, which in turn activates the effectors, the extracellular signal-regulated kinases ERK1 and ERK2. MEK1/2, as well as ERK1/2, are expressed at relatively high levels compared with the Raf isoforms in cardiomyocytes. Since the expression level of MEK1/2 increases only slightly in both models, their expression level is probably sufficient to maintain the requirements for remodeling myocytes (Figure 5). In contrast, ERK1 and ERK2 increased markedly in both models and were more pronounced during fetal reprogramming. ERK3 reveals 43% homology to ERK1/2 and appears as a 63 kDa isoform in morphogen-stimulated cardiomyocytes (Figure 5).

### 2.8. Structural and MAPK Remodeling Are Observed in Cardiomyocytes of Patients with Dilated Cardiomyopathy

Primary cultures of adult cardiomyocytes offer an ideal toolbox to analyze the general behavior of stress-activated myocytes in the diseased myocardium since they transfer their genetic, epigenetic, and proteomic statuses into the culture dish [25,27]. By this a number of novel biomarkers of cardiac remodeling and failure can be determined and basic principles of cardiac disease development analyzed [24,25,27]. Here, we wanted to know whether structural and MAPK remodeling can be observed in cardiomyocytes of patients with dilated cardiomyopathy (DCM). Expression of α-actinin-1-positive cardiomyocytes can be identified either singularized in the fibrotic area or in the cell collective of patients with DCM (Figure 6A). Phosphorylated ERK1/2 indicates that MAPK signaling is still intact. Furthermore, the expression of BNP in α-actinin-1-positive cardiomyocytes is in agreement with increased ANP and B-Raf levels in the myocardium of patients with DCM [11]. Reexpression of ERK3 in a number of α-actinin-1 positive myocytes indicates that structural remodeling is connected to MAPK reprogramming (Figure 6A). Unloading similar to unstimulated cultured cardiomyocytes might also occur in patients with left ventricular assist device (Figure 6B). The amount of activated ERK (P-ERK1/2) continously decreases from the right ventricle (RV) over the septum (Sep) to the left ventricle (LV). An age-matched control is shown (Control).

## 3. Discussion

Here we demonstrate that reprogramming of the MEK/ERK module is part of cardiomyocyte remodeling (summarized in Figure 7). Serum induced hypertrophic remodeling while treatment of cardiomyocytes with endothelial morphogens induced fetal remodeling. Both processes appeared to be independently evoked by circulating (serum) or secreted cardiac endothelial cell-derived mediators (morphogen). Activation of the MAPK induced fetal remodeling and ischemic resistance while hypertrophic remodeling was not blocked by the MEK1/2 inhibitor UO126. Since fetal remodeling is generally regarded as part of the hypertrophic process, this distinction implies opposing therapeutic consequences when targeting the MAPK. The activation of the Ras/Raf/MEK/ERK pathway might be beneficial in an ischemic challenged heart, but detrimental in the myocardium with signs of strongly dedifferentiated cardiomyocytes. To determine the quality of the Ras/Raf/MEK/ERK pathway, we analyzed expression and reexpression of its components.

All classical Ras isoforms were present in adult cardiomyocytes but at different expression levels; H-Ras and K-Ras are more abundant than N-Ras. We assume that slight increases of H-Ras in both models indicate an adaptive response of the cell to an increased protein turnover, rather than a contribution to specific growth effects. The mild involvement in cardiomyocyte remodeling is supported by data obtained with transgenic mice overexpressing cardiac-targeted V12H-Ras. H-Ras promoted only left ventricular hypertrophy in homozygotic but not heterozygotic mice, despite significant expression of the oncogenic V12H-Ras [10]. In contrast, K-Ras was downregulated in remodeling cardiomyocytes and was more pronounced in dedifferentiating cultures than hypertrophic cultures; an unexpected finding since K-Ras-targeted mice show fragile ventricular walls and die at an embryonic stage [33,34]. An explanation for this discrepancy might be that K-Ras is needed for the cycling of myocytes but may play only a minor role in terminally differentiated adult cardiomyocytes which do not proliferate. N-Ras seems to substitute K-Ras since there is an inverse correlation in the relative amounts of both isoforms in cardiomyocytes. It might be that N-Ras and H-Ras have partial functional overlaps with K-Ras since H-Ras- and N-Ras-deficient mice grow normally [33].

Regulation of Rap1 and Rap2 adds to the complexity of Ras-dependent signaling in remodeling cardiomyocytes. Rap1 is thought to be involved in the transition from hypertrophy to heart failure [4]. As there are essential differences between Ras and Rap substrates, qualitative differences between Ras and Rap signaling are also to be expected [35]. Surprisingly, Rap1, the closest relative of Ras and usually undetectable in cardiomyocytes, became massively reexpressed during fetal remodeling and only slightly increased after hypertrophic remodeling. This event is insofar remarkable since Rap1 has been identified as a revertant of the transformed phenotype of fibroblasts carrying a mutated K-Ras and shares extensive homology within the regions involved in effector binding as well as in the K-Ras effector domain [30]. Rap1-deficient mice show partial embryonic lethality and bleeding [36]. During fetal remodeling, the level of Rap1 appears to be inversely related to the loss of K-Ras. Whether this switch from K-Ras to a functional Rap1 leads to an effector machinery responsible for a MAPK dependent fetal remodeling of cardiomyocytes is an important mechanistic issue but requires clarification.

In addition to the switch from H-Ras/K-Ras protein expressions to H-Ras/N-Ras in hypertrophic cardiomyocytes and H-Ras/N-Ras/Rap1 in fetal remodeling myocytes, the complexity of MAPK still increases through the reprogramming of direct effectors of Ras signaling. The first mammalian and most intensively studied targets of Ras were the Raf serine/threonine kinases. Raf proteins appear to exert non-redundant functions since B- and C-Raf knock-out die in utero while A-Raf deletion causes early postnatal death [37]. A-Raf and C-Raf are significantly expressed in the heart, whereas in carefully conducted studies, B-Raf proteins (in contrast to their RNA) were detectable only at extremely low levels in the cardiac tissue [38,39,40]. Strong increases of B-Raf in the failing heart has been demonstrated in patients with dilated cardiomyopathy as well as in animal disease models, being in line with an increased MAPK activation [11,13].

In contrast to cardiac tissue, B-Raf was not detected in isolated rat neonatal nor in adult cardiomyocytes [39,41]. B-Raf protein expression first became apparent in cultured cardiomyocytes after stimulation with oncostatin M, an extremely potent remodeling factor [11]. Even when the 95 kDa isoform is barely detectable on the protein level, B-Raf is considered as the major MEK1/2 activator due to a 50-fold higher biochemical activity than C-Raf [29,30]. There is accumulating evidence that A-Raf is a minor activator of MEK1/2, pointing again to B-Raf as the major MAPK activator [29,42]. All three Raf isoforms are expressed in cultured adult cardiomyocytes but under different growth stimulants and at various expression levels. The 95 kDa full-length B-Raf was barely detectable in controls but became significantly expressed during hypertrophic and fetal remodeling. The level of B-Raf was higher and appeared earlier in the morphogen-treated group than in serum-stimulated cultures. Knock-down of B-Raf by siRNA drastically reduced MAPK cascade activation and dedifferentiation, suggesting that B-Raf is strongly involved in fetal remodeling. Furthermore, the sharp increase of B-Raf in serum stimulated cultures indicates a further role in hypertrophic remodeling which might be involved in MAPK cascade-independent cascades. Evidence for autonomous B-Raf functions comes from in vitro and in vivo forelimb studies showing that B-Raf mediates muscle precursor cell migration independently from ERK1/2 phosphorylation [15].

B-Raf and C-Raf are proposed to bind Rap1 in contrast to A-Raf. However, it has been suggested that Rap1 activates B-Raf but does not activate C-Raf [30]. If this is also the case in myocytes, the Rap1/B-Raf interaction regarded as specific for neuronal cells [43] could similarly exist in dedifferentiating cardiomyocytes. As cAMP might be involved in the activation of Rap1 and cardiomyocyte lengthening shows morphological similarities to neurite outgrowth, we hypothesize that Rap1/B-Raf signaling, dedifferentiation, and cell lengthening could employ the same mechanistic principles as neurite outgrowth in PC12 cells.

A-Raf is lost during fetal remodeling but strongly increased in hypertrophic cardiomyocytes. It has been hypothesized that cells requiring a high level of metabolic activity express A-Raf [40]. Pyruvate kinase M2 has been reported to bind A-Raf but not C-Raf or B-Raf, and this interaction is proposed to lead to an active glycolytic enzyme [44]. It is well known that during cardiac hypertrophy a shift occurs from oxidation of fatty acids to a greater dependence on carbohydrates [45]. Pathophysiological conditions such as chronic ischemia, hypertrophy, and failure are associated with a marked shift from fatty acid to glucose utilization. It is thought that the hypertrophic state rather than the enhanced workload is the ultimate cause of this shift toward glucose use and is extensively discussed by van Bilsen et al. [45]. A-Raf may, therefore, be a potential mediator of metabolic remodeling combined with changes in energy use in hypertrophic cardiomyocytes. Vice versa, the loss of A-Raf in dedifferentiating cells could initially be an adaptation to ischemia; however, the reduced metabolic capacity might not be compensated in the long term by the other two Raf isoforms, resulting in cardiomyocyte failure.

At the protein levels of MEK1/2 and ERK1/2, comparatively little change occurred in the Ras/Raf/MEK/ERK cascade. Only slight elevations in MEK1/2 were observed as expression levels might be sufficient to transmit activating signals to ERK1/2. The increases in total ERK1 and ERK2 as “workhorses” likely reflect the need of cardiomyocytes to sustain the level of signaling cues during fetal remodeling. Surprisingly, the atypical ERK3, which is barely detectable in the control group, was reexpressed during fetal remodeling. As we detected the 63 kDa but not the 90 kDa isoform, we conclude that the observed band is a truncated version of ERK3 [46]. We do not have any information about the function of ERK3 in adult cardiomyocytes but targeted disruption of this kinase results in early postnatal death of mice, with markedly reduced weights of lung and heart, indicating that ERK3 is necessary for fetal cardiac development [47].

## 4. Materials and Methods

### 4.1. Experimental Design and Settings

Stimulation of cardiomyocytes with serum represents the model of hypertrophic remodeling since these cultures accumulate protein. Treatment with endothelial morphogens serves as a model of fetal remodeling which is characterized by atrophy and reexpression of fetal genes. Morphogens (Morpho) were obtained by conditioning of the supernatant (Medium 199) through cardiac microvascular endothelial cells, as previously described [28]. This supernatant was not externally supplemented by any growth stimulants such as insulin or FGFs. Only cardiomyocyte cultures were used, which show several criteria of cell viability such as absence of apoptosis or blebbing, the ability to dedifferentiate and to form new sarcomeres as well as general benchmarks of growth (protein synthesis, cell spreading, etc.).

Cardiomyocytes were isolated from 6 independent perfusions of adult rat cardiomyocytes (*n* = 6) unless stated otherwise. Statistical relevance is indicated in the figure legends. Cells were plated at high density (1.5 × 10^4^ cells/cm^2^) or at low density for ischemic experiments and for the determination of cell length and diameter (0.5 × 10^4^ cells/cm^2^). Each isolation was divided into 5 groups and treated as indicated after a recovery period of one day. Additional groups (6 to 9) were analyzed for a more detailed sudy of remodeling effects:Control (Con 0 d), day of stimulation start;Myocytes stimulated for 10 days with 5% serum (Serum);Myocytes stimulated for 10 days with endothelial morphogens (Morpho);Myocytes stimulated for 20 days with 5% serum (Serum);Myocytes stimulated for 20 days with endothelial morphogens (Morpho);In some experiments, we use a new control group of cultured cardiomyocytes in the absence of any stimulants for 6 days (unloaded; Con 6 d);To determine ischemic resistance, cultures were pretreated for 6 days and kept for 9 h at 1% O2 in glucose-free PBS;To determine MAPK-dependent fetal remodeling and cell size parameters, cultures were treated either for 10 min or 6 days;In order to determine the effect of B-Raf on MAPK activation and fetal remodeling, cardiomyocytes were pretreated for 3 days with control or B-Raf siRNA and then stimulated with morphogens.

While untreated controls at days 10 and 20 are highly desirable, it is not possible to maintain primary cultures of adult cardiomyocytes without any growth supplements such as insulin for extended culture periods as they increasingly start to die from day 7. Our intention was not only to analyze differences between hypertrophic and fetal remodeling, but also to differentiate the involved recomposition of proteins from remodeling cardiomyocytes with an expression pattern comparable to the in vivo status (Con 0 d). Note that terminally differentiated adult cardiomyocytes do not proliferate.

### 4.2. Culture of Adult Cardiomyocytes, Basic Medium, and Preparation of Endothelial Morphogens

Ventricular cardiac myocytes of 2–3-month-old male Wistar rats were isolated, plated (1.5 × 10^4^ cells/cm^2^), and cultured in basic medium as previously described [48]. Basic medium consists of Medium 199 with Earle’s balanced salts (Sigma-Aldrich, St. Louis, MO, USA) without L-glutamine, including 25 mM HEPES, 25 mM NaHCO_3_, 100 IU/mL penicillin, and 100 μg/mL streptomycin, and supplemented with 2 mM L-carnitine, 5 mM creatine, and 5 mM taurine (Sigma-Aldrich, St. Louis, MO, USA). All cardiomyocyte cultures were treated continuously with 10 μM 1-β-D-arabinofuranosyl cytosine (Sigma-Aldrich, St. Louis, MO, USA) to prevent nonmyocyte growth. To avoid the protective effect of cell–cell contacts, cardiomyocytes were plated at low density for ischemic experiments (0.5 × 10^4^ cells/cm^2^). Cardiac microvascular endothelial cells were isolated, cultured, and characterized as previously described [49]. Confluent monolayers of endothelial cells were washed twice with PBS and cultured for two days in basic medium [28]. The conditioned basic medium (morphogen, morpho) was diluted with basic medium in a 4:1 (vol/vol) ratio. Cardiomyocytes were allowed to recover for one day before experiments began. Media were replaced every other day.

### 4.3. Determination of the Protein Content, Cell Size Parameters, and siRNA Knock-Down

We examined the effect of Morpho and Serum on protein accumulation by determination of the protein-to-DNA ratio. The amount of DNA is considered as a reliable measure of the relative cell number in adult cardiomyocytes since there is no change in DNA content during adult growth. The corresponding 6-day-old culture served as 100% reference. At the end of the experiments cultures were washed twice with Hanks’ balanced salt solution, fixed for 1 h in 10% TCA, washed twice with 10% TCA, thrice with 95% ethanol, then air dried and extracted in 0.3 M NaOH. Aliquots were removed for measuring total protein content by the detergent compatible protein assay (Bio-Rad, Hercules, CA, USA). In order to measure the DNA content the pH of the lysate was adjusted to 12.3 with 10 mM EDTA and determination was carried out as described elsewhere [50]. Protein content was determined as previously described [28]. Images with a phase-contrast microscope (Leica, Wetzlar, Hesse, Germany) were obtained using a digital camera (Leica, Wetzlar, Hesse, Germany) and additional software cell length as well as diameter of cardiomyocytes at low density were determined with the image J software as previously described [51]. B-Raf siRNA (ON-TARGETplusSMARTpool L-094802, XM_231692) was obtained from Dharmacon (Lafayette, CO, USA) together with the corresponding transfection kit. Transfection of cardiomyocytes was performed using 0.5 µM siRNA per culture as previously described [11]. Experiments started after three days of siRNA knock-down. Five micromolar UO126 was added to the cultures 1 h prior to stimulation. Isoproterenol was obtained from Sigma-Aldrich (St. Louis, MO, USA).

### 4.4. Western Blot Analysis and Microscopy

Fluorescence microscopy was performed as previously described [52]. For Western blot analysis, cell cultures were lysed in 5-fold concentrated Laemmli gel running buffer after three washes with PBS. Protein determination was performed with the detergent compatible protein assay from Bio-Rad (Hercules, CA, USA). For high resolution, samples were heated at 70 °C for 10 min in 30% power buffer (0.1 M tromethamine-hydrochloride, 10% sodium dodecyl sulfate (SDS), 10 mM ethylenediaminetetraacetic acid, 0.15 M dithiothreitol; pH 8.0). Insoluble material was removed by centrifugation at 14,000× *g* for 2 min. Total protein per lane (10 µg) was applied to a 4–12% SDS polyacrylamide gradient gel and samples were blotted onto nitrocellulose (all Invitrogen, Heidelberg, BW, Germany). Protein transfer was monitored with RedAlert (Merck, Darmstadt, Hesse, Germany). Membranes were treated with Signal Enhancer (Perbio Science, Erembodegen, Belgium) and immunoreactive proteins were visualized with corresponding HRP-conjugated secondary antibodies on Hyperfilm (GE Healthcare Life Sciences, Freiburg, BW, Germany) using the Super Signal Pico for structural proteins and the Femto Detection Kit (Perbio Science, Erembodegen, Belgium) for signaling pathway proteins. The following antibodies were used. H-Ras (clone 18), A-Raf (clone 1), B-Raf (clone 13), C-Raf (clone 53), Rap1 (clone 3), Rap2 (clone 12), calsarcin 1 (clone 41), MEK1 (clone 25), MEK2 (clone 96), Erk1 (clone MK12), Erk2 (clone 33), and Erk3 (clone 30) were from BD Biosciences (Palo Alto, CA, USA). N-Ras (sc-31) and K-Ras (sc-30; recognizes K-Ras-2A and 2B) were from Santa Cruz Biotechnology (Santa Cruz, CA, USA). Vinculin (clone VIN-11-5), α-tubulin (clone DM1A), desmin (clone DE-U-10), β-actin (clone AC-15), and α-smooth muscle actin (clone 1A4) were purchased from Sigma-Aldrich (St. Louis, MO, USA). Skeletal actin (clone O.N.4) was from US Biological (BIOZOL Diagnostica, Leipzig, SN, Germany) and muscle actin (HHF35) from Biocare Medical (ZYTOMED Systems, Berlin, BE, Germany). Pan-actin antibody (Clone C4; reacts with all 6 actin isoforms) was purchased from Boehringer Mannheim (Mannheim, BW, Germany). Blots were scanned with a STORM 860 (Molecular Dynamics, Freiburg, BW, Germany) and analyzed by ImageQuant software (Molecular Dynamics, Freiburg, BW, Germany). P-ERK1/2 (D13.14.4E) and total α-actinin (Cat Nr. 3134) was from Cell Signaling (Danvers, MA, USA) and α-actinin 1 from Epitomics (EP2527Y; Abcam, Cambridge, MA, USA). BNP (8D5B4C11) was purchased from Abcam (Cambridge, MA, USA).

### 4.5. Statistical Analysis

Statistical analysis was performed with the Graph Pad Prism (GraphPad Software, San Diego, CA, USA) using the Student’s *t*-test. *p*-values of <0.05 were taken as statistically significant.

## 5. Conclusions

In conclusion, we showed a correlation between changes in the structural proteome with reprogramming of the signaling proteome. We discussed potential relationships within the Ras/Rap/Raf transduction network and are aware of the study limitations by neglecting other signaling pathways. We demonstrated that B-Raf induces fetal remodeling through the activation of the MAPK pathway. Knock-down of B-Raf sharply reduced phosphorylation of MEK1/2 and ERK1/2 and reexpression of SM-actin and actinin-1. A limitation of our study is that we do not know how far our results can be extrapolated to clinical practice as this requires large cohorts of patients to be analyzed. Nevertheless, it is imaginable that different drug types at certain stages of remodeling (progression) will automatically have a disparate impact on the outcome; therapies will either be beneficial or detrimental to cardiomyocytes due to their ability to respond to, or block, a signaling input. Reorganization of the signaling proteome should also affect the susceptibility of cardiomyocytes to mechanical stress as one of the major stimuli triggering heart failure. Finally, anti-cancer drugs might also influence the progression of cardiac diseases.

## Figures and Tables

**Figure 1 ijms-21-06348-f001:**
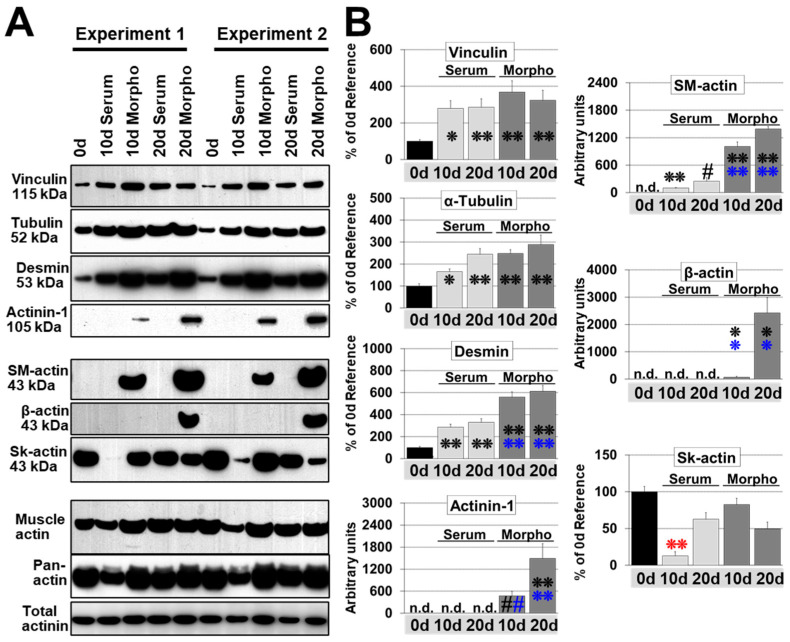
Fetal but not hypertrophic remodeling leads to the reexpression of fetal genes. Primary cultures of adult cardiomyocytes were harvested at the start of the experiment (0 d, reference), after 10 (10 d), and 20 days (20 d) for western blot analysis. Zero-day cultures reflect the protein composition of cardiomyocytes in the normal adult heart. Cultures were stimulated with 5% serum (Serum) and conditioned supernatant of cardiac microvascular endothelial cells containing no growth supplements (Morpho, 80%). (**A**) Total protein (10 µg) were analyzed for the expression of vinculin, α-tubulin (Tubulin), desmin, skeletal actin (Skel-actin), and the reexpression of α-actinin-1 (Actinin-1), β-actin and α-smooth muscle actin (SM-actin). Muscle actin and pan-actin served as loading controls. Detected molecular sizes are indicated. (**B**) Quantitative evaluation and statistical analysis of (**A**). Note that the order of groups is presented differently from (**A**). Single and double stars indicate *p* < 0.01 and *p* < 0.001, respectively. Hash signs indicate *p* < 0.05. Black signs refer statistically to Con 0 d, blue signs to serum, and the red to all other groups.

**Figure 2 ijms-21-06348-f002:**
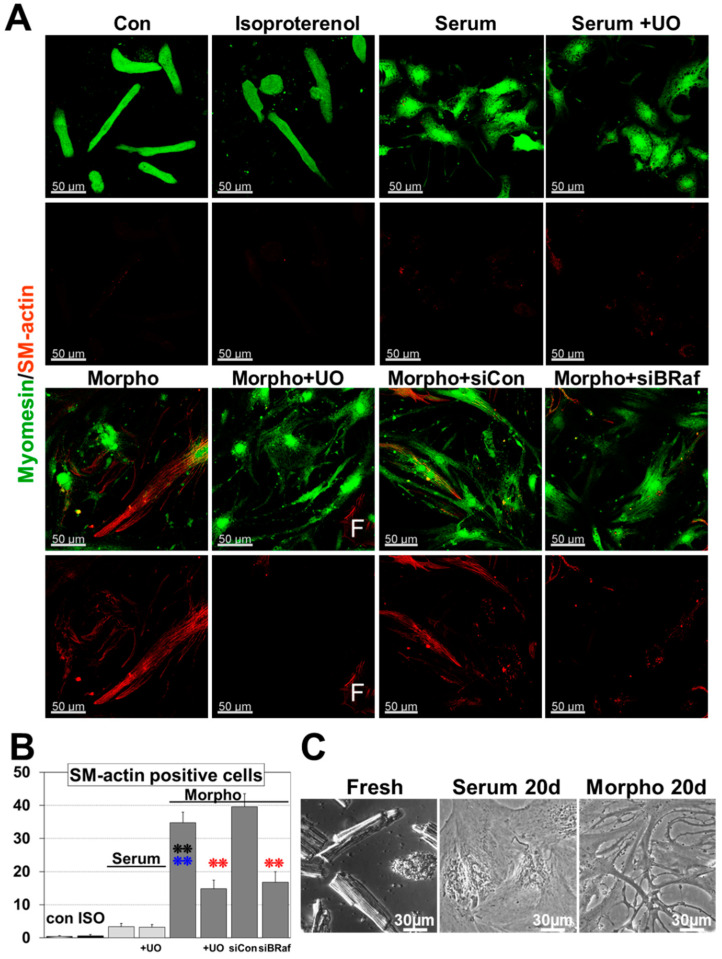
Morphogen but not serum induces Ras/Raf/MEK/ERK cascade (MAPK)-dependent remodeling and dedifferentiation. (**A**) Myomesin and α-smooth muscle actin (SM-actin) fluorescence images after 6-day treatment. SM-actin serves as marker for dedifferentiation/fetal remodeling. Primary cultures of adult cardiomyocytes were untreated or treated with 20 nM isoproterenol (ISO), serum (Serum) and morphogens (Morpho). UO126 (5 µM) was added as an inhibitor of the MAPK pathway (+UO) 1h before experimental start. Note that knock-down of B-Raf (siBRaf) was performed for three days prior to stimulation. Single and double stars indicate *p* < 0.01 and *p* < 0.001, respectively. Black signs refer statistically to Con, blue signs to serum, and red compares Morpho (Morpho, Morpho siCon) statistically with and without UO/siB-Raf (+UO, siBRaf). A SM-actin positive fibroblast acts as positive control and is indicated by a white F. (**B**) Quantitative evaluation of the number of SM-actin positive cells and statistical analysis of (**A**). (**C**) Phase-contrast images of freshly isolated cells (Fresh), as well as Serum- (Serum 20 d) and Morpho-treated (Morpho 20 d) cardiomyocytes after 20 d showing long-term effects.

**Figure 3 ijms-21-06348-f003:**
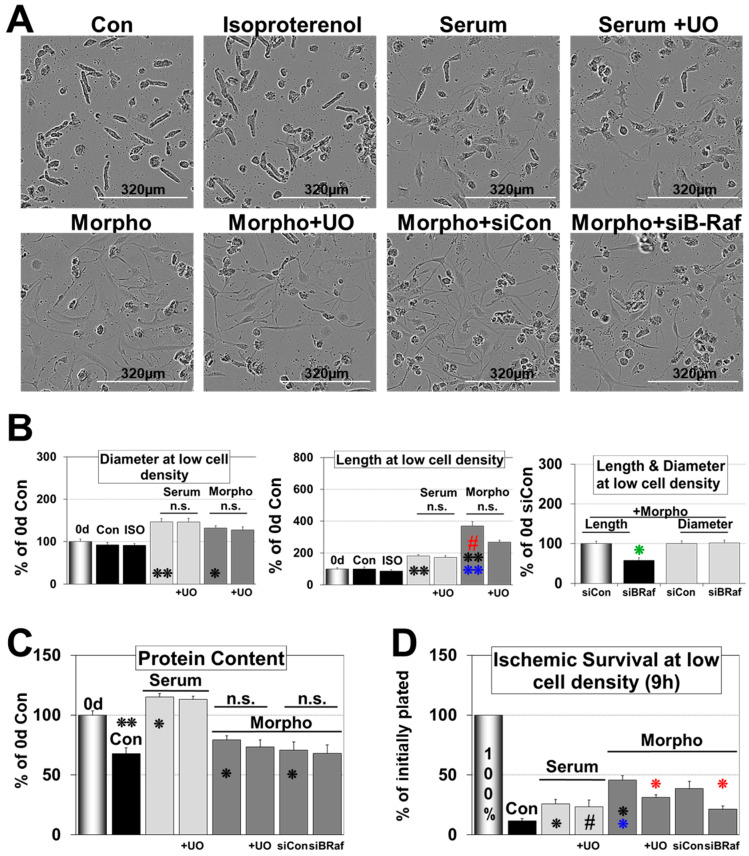
Fetal remodeled but not hypertrophic cardiomyocytes show MAPK dependent ischemic resistance. Primary cultures of adult cardiomyocytes were untreated (Con) or treated with isoproterenol (ISO), serum (Serum), and morphogens (Morpho) for 6 days. Cardiomyocytes were seeded at low density except in (**C**). UO126 (5 µM) was added (+UO) 1 h before experimental start. Note that cultures were pretreated with siRNA (siCon, siBRaf) three days before experimental start. Single and double stars indicate *p* < 0.01 and *p* < 0.001, respectively. Hash signs indicate *p* < 0.05. Black signs refer statistically to the 0 d group, blue signs to serum, and red compares Morpho (Morpho, Morpho siCon) statistically with and without UO/siB-Raf (+UO, siBRaf). The green star refers to the siCon group. (**A**) Phase-contrast images of cardiomyocyte cultures. (**B**) Quantitative and statistical evaluation of cell length and diameter of (**A**). (**C**) Total protein content of 6 d-treated cultures at high cell density were compared with cardiomyocytes at experimental start (0 d). (**D**) Cardiomyocytes were pretreated for 6 days with 0.2 mg/mL albumin (Con), serum, serum + UO126, morphogen, morphogen + UO126. At the end of a 6-day culture period, all cells were counted (reference value 100%) and then kept in glucose-free PBS at 1% O_2_. After 9 h of ischemia, cardiomyocytes were fixed in 4% paraformaldehyde and recounted.

**Figure 4 ijms-21-06348-f004:**
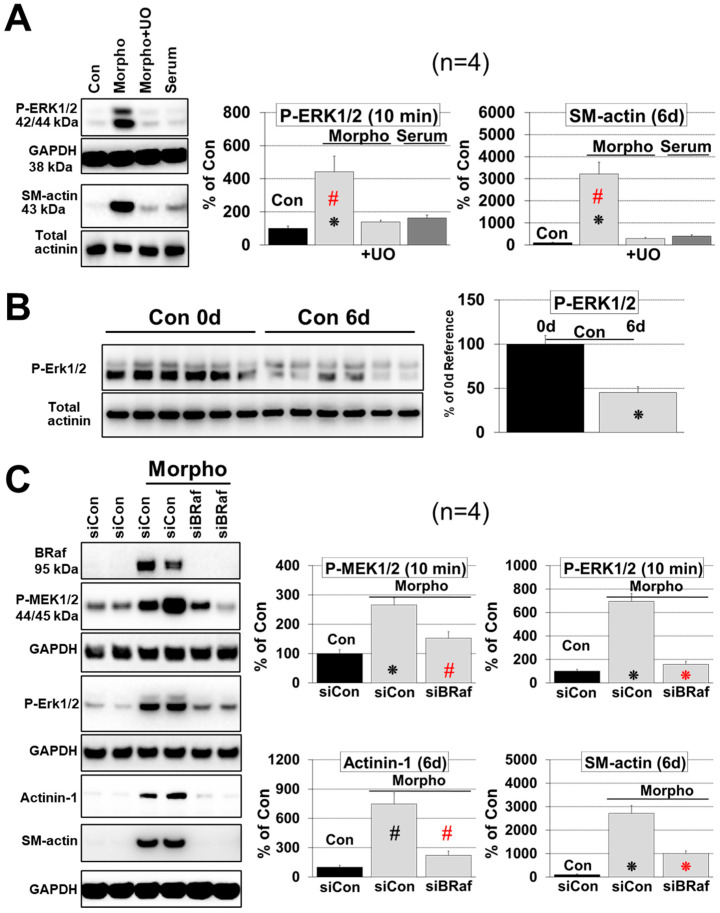
Knock-down of B-Raf and treatment with UO126 (+UO) abrogates MAPK cascade activation and fetal remodeling. Isolated cardiomyocytes (*n* = 4) were allowed to recover for 1 day (Con 0 d) and then treated for 10 min (10 min) or 6 days (6 d) for western blot analysis. Cultures were treated with basic medium (Con), with serum (Serum), or with morphogens (Morpho) as indicated. MEK1/2 inhibitor UO126 (UO, 5 µM) was added 1 h before the experimental start as indicated. Quantitative evaluation and statistical analysis of western blots (WB) are shown. Single star and hash sign indicate *p* < 0.01 and *p* < 0.05, respectively. Black signs refer statistically to Con. Red signs compare Morpho (Morpho, Morpho siCon) statistically with and without UO/siB-Raf (+UO, siBRaf). (**A**) WB analysis of ERK1/2 (P-ERK1/2) activation and dedifferentiation (SM-actin) in the absence or presence of UO (+UO). (**B**) WB analysis of P-ERK1/2 of untreated cardiomyocytes (cultured in basic medium) at 0 d and after 6 d (*n* = 6). (**C**) After a three-day pretreatment period with siRNA, cardiomyocytes were cultured in basic medium or stimulated with Serum and Morpho for WB of α-actinin-1 (Actinin-1) and α-smooth muscle actin (SM-actin) for six days. In order to determine MEK1/2 and ERK1/2 phosphorylation (P-MEK1/2, P-ERK1/2) cultures were made quiescent after six days for one day in basic medium and then treated as indicated for 10 min.

**Figure 5 ijms-21-06348-f005:**
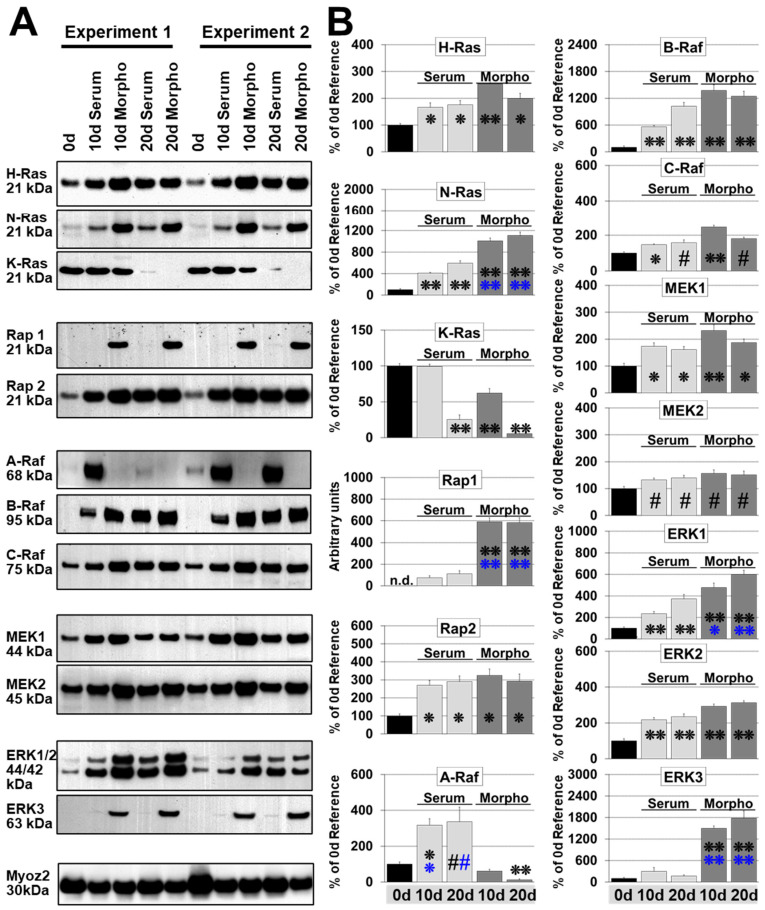
MAPK remodeling in dedifferentiating cardiomyocytes is distinct from hypertrophic myocytes. Primary cultures of adult cardiomyocytes were harvested at the experimental start (0 d, reference), and after 10 (10 d), and 20 days (20 d) for western blot analysis. Cultures were stimulated with 5% serum (Serum) and conditioned supernatant (Morpho, 80%) of cardiac microvascular endothelial cells containing no growth supplements. Single and double stars indicate *p* < 0.01 and *p* < 0.001, respectively. Black stars refer statistically to Con, blue stars to serum. Hash signs indicate *p* < 0.05. For A-Raf blue symbols refer statistically to morphogens. (**A**) Total protein (10 µg) was analyzed for the expression of components of the Ras/Raf/MEK/ERK pathway, as well as the Ras-related proteins Rap1 and Rap2. MYOZ is Calsarcin-1 and serves as a loading control. Note that we detected Erk3 as the 63 kDa form but not as full length 90 kDa protein. Detected molecular sizes are indicated. (**B**) Quantitative evaluation and statistical analysis of (**A**). Note that the order of groups is presented differently from (**A**).

**Figure 6 ijms-21-06348-f006:**
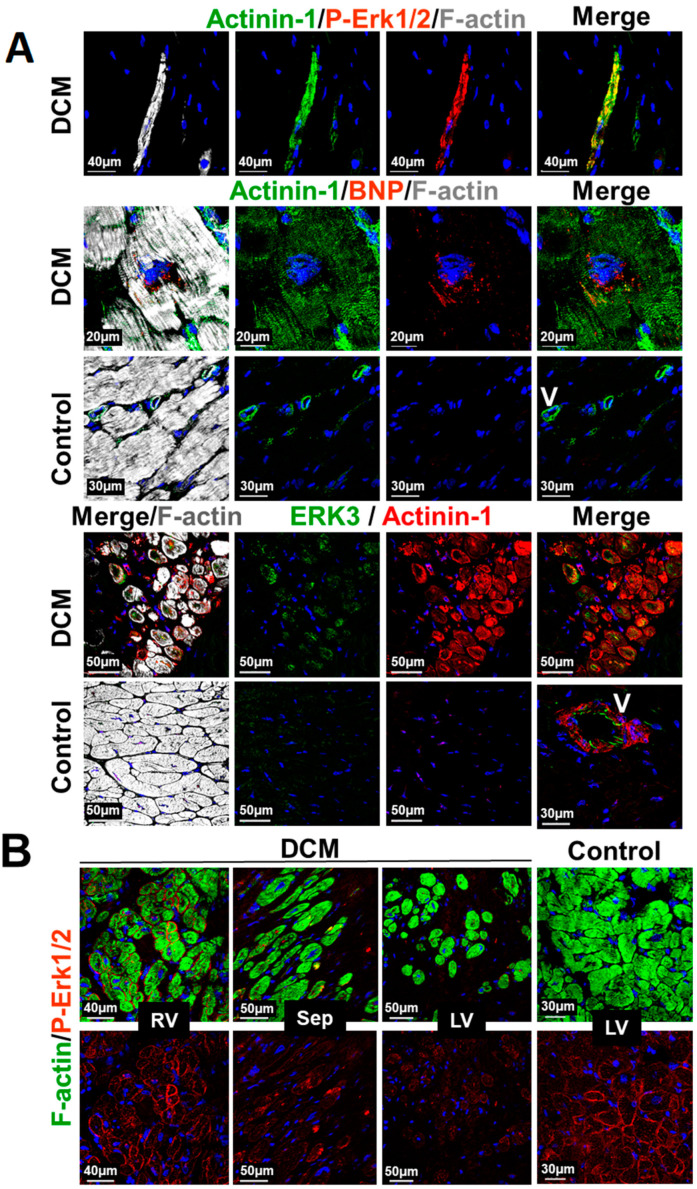
Structural and MAPK remodeling of cardiomyocytes in the myocardium of patients with dilated cardiomyopathy. (**A**) Confocal images of a transplanted patient with end-stage dilated cardiomyopathy (DCM). Upper images show an α-actinin-1 (Actinin-1)- and P-Erk-1/2-positive cell isolated from other cardiomyocytes in a fibrotic area of the myocardium. P-ERK1/2 indicates that the MAPK is still intact. Middle images depict α-actinin-1 positive cardiomyocyte expressing BNP. Lower images show cardiomyocytes reexpressing α-actinin-1 as well as Erk3. Controls show neither ERK3 nor actinin-1 expression. V indicates a vessel and serves as positive control for actinin-1. (**B**) Confocal fluorescence images of a transplanted 12-year-old patient with left ventricular assist device. Note the P-ERK1/2-decreasing gradient from the right ventricle (RV) over the septum (Sep) to the left ventricle (LV). LV control serves an age-matched patient with Tetralogy of Fallot.

**Figure 7 ijms-21-06348-f007:**
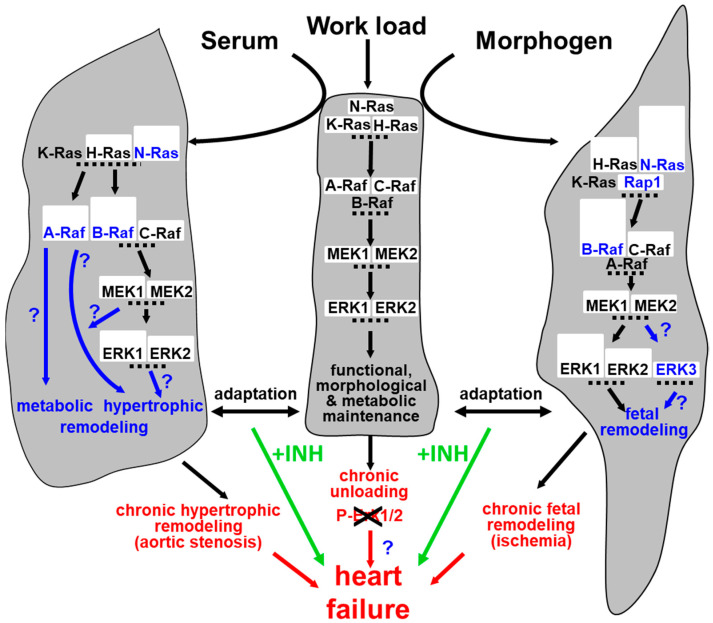
Remodeling of the MAPK pathway in the hypothetical context of cardiac adaptation and failure. The myocardium has a certain capacity to adapt to increased load either by hypertrophy (e.g., evoked though aortic stenosis) or by fetal remodeling (e.g., evoked through ischemia). However, when cardiac demands are not met and become chronic, heart failure may develop. Cardiomyocyte remodeling does not only result in structural reorganization but also involves reprogramming of the MAPK. A third road to heart failure might be produced by cardiac unloading due to the absence of appropriate stimuli rendering cardiomyocytes unfunctional. The interference by MAPK inhibitors (+INH) might disturb adaptive processes and enhance the development of heart failure.

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
