# Peer review of "The MEK/ERK Module Is Reprogrammed in Remodeling Adult Cardiomyocytes"

_ijms, 2020, doi:10.3390/ijms21176348_

Round 1
Reviewer 1 Report
Thank you very much for the opportunity to review the manuscript submitted by Dr. Manfred and colleagues to the International Journal of Medical Sciences entitled "The MEK/ERK module is reprogrammed in remodeling adult cardiomyocytes." This is an important topic in the understanding of the development of heart failure and I read the manuscript with much interest. Unfortunately, the manuscript requires major structural changes to improve its clarity. In particular:
- Section 2.1 in the results termed "experimental settings" should be moved to methods
- The authors never state the method by which total protein is determined
- There are several figure panels that are not mentioned at all in the text of the results, for example, the pERK1/2 immunofluorescence experiment in Figure 1C, the HRAS, NRAS, KRAS RAP1 and RAP2 blots in Figure 4 A and B
- Several figure panels appear to be grouped with the incorrect figure. For example, panel 2E should be in figure 3 and panel 5A should be in figure 4
- The authors miss an opportunity to discuss the large body of literature concerning the pathogenesis of cardiac hypertrophy in the RASopathies, genetic disorders caused by germline pathogenic variations in the RAS/MAPK pathway
- The authors mention that they plan to evaluate genetic manipulation of expression of components of the MAPK pathway as well as phosphorylation states of these proteins in the serum and morpho conditions, however, unfortunately, these experiments must be performed for the study to be meaningful
Reviewer 2 Report
In this manuscript, Kubin Thomas and colleagues described the different responses of adult rat cardiomyocytes towards hypertrophic treatment (by serum) and fetal remodeling (by morphogens from conditioned medium). The authors further illustrated that morphogens potentially activated MAPK pathway through enhanced expression of phosphor-ERK1/2, Rap1 and Erk3. IHCs using human diseased heart samples supported the findings. Based on these observations, the authors proposed that MAPK signaling cascade may play different roles in cardiac hypertrophy and cardiomyocyte dedifferentiation. It has been well-known that activation of fetal gene program is one of the markers for pathological cardiac hypertrophy. This research proposed somehow different idea that the fetal gene program is partially distinct from cardiac hypertrophy. Nevertheless, the study is not designed/performed rigorous enough and suffer from the following issues.
Major issue:
- The authors compared all treated samples (cultured for 10-20 days) to freshly isolated cardiomyocyte (control 0d), causing the major caveat of the study. The unstimulated cells cultured in basal media should be collected the same as the treated samples, and these cells should be the real control. Otherwise, how could the authors exclude the possibility of culture effect? This caveat reduced the enthusiasm of this manuscript.
- Serum treatment has been used to promote neonatal cardiomyocyte hypertrophy. Here in this research the authors mainly used adult rat cardiomyocyte as study materials. The authors should show the extend of hypertrophy that this serum treatment could lead to. It is possible that the serum treatment is not strong enough as morphogen treatment, so that the authors failed to observe activation of MAPK signaling. Supporting this notion, the cells treated with morphogen for 10d shown in Fig.2B exhibited larger cell size than the that treated with serum.
- The authors aimed to study the role of Ras/Raf/MEK/ERK pathway “through culture-induced hypertrophic or fetal remodeling”. Cardiac hypertrophy could also be induced by pharmacological molecules such as phenylephrine and isoproterenol. The authors should use at least one drug-induced disease model to have broader appealing of their research.
- Fig1A, the 0d sample seemed to have lower amount of loading protein (based on the pan-actin) compared with other samples, which reduced the reliability of the quantification data. Additionally, as the cultured cardiomyocytes changed cytoskeleton machinery during treatment, it may not be appropriate to use actin as loading control. Instead, other housekeeping genes, such as GAPDH, are encouraged to be used.
- Fig2B, quantification analysis is missing to show the percentage of cells that were undergoing remodeling. Meanwhile, immunofluorescent staining and quantification of control cells (cultured in basal medium) should be included and compared with.
- The authors used diseased heart samples to show the correlation of ERK activation with fetal program. However, similar staining data from normal heart were not provided. It’s difficult to make any conclusion based on current IHC results. In addition, in hypertrophic sections, marker gene expression, such as ANP and BNP should be evaluated to demonstrate the pathological condition of the samples.
- The tittles for Y axis on all the histogram plots through the manuscript were missing.
Reviewer 3 Report
I was pleased to review the manuscript proposed by Thomas and colleagues who investigated the MAPK cascade modulation in fetal and hypertrophic remodeling cardiomyocytes. The authors extensively characterized the MAPK pathway in adult rat cardiomyocytes first and then extended some investigation to some human cardiomyocytes of failing myocardium. The authors demonstrated that fetal and hypertrophic remodeling occurs through reprogramming of MAPK. The study is well characterized by a biochemical standpoint. I suggest the following minor revisions:
- Material and Method section lacks in the description of patient-derived cardiomyocytes isolation. Please, provide this information.
- Patient-derived cardiomyocytes (Fig. 1C, 3C, 5A) have been characterized by confocal analysis. Together with representative images, it would be helpful to report a statistic abut the percentage of cardiomyocytes that stain positive for the proteins of interest. This information can be extrapolated by analyzing the images and counting the positive cells/total cells.
- Based on what reported in Fig. 2 legend, cell viability has been accessed just counting the cells. More accurate and reliable viability methods should be used, such as trypan blue to exclude dead cells. Please, provide additional information about how viability has been performed and please address this point accordingly.
Reviewer 4 Report
This manuscript looks as the MAPK signaling pathway during cardiomyocyte remodeling. The work combines in vitro studies with analyses of human tissues. The study demonstrates correlations but does not address mechanisms with use of knockdowns or specific inhibitors.
- The contrast for the western blot data was too high. This should be reduced to more accurately reflect data. Similarly the immuno data was overexposed
- The appropriate loading control for p-ERK is not actinin but total ERK (Figure 3)
- Last two lines of the abstract should be removed. The implications for cancer treatment are a large stretch from the current study.
- What is meant by cardiac regeneration in line 55 of the introduction? There is very limited cardiomyocyte proliferation in adult hearts. This should be clarified.
- The list of conditions in lines 102-11 should be in the figure legends not in the text
Round 2
Reviewer 2 Report
In this revised manuscript, the authors have addressed most of the concerns from this reviewer. These efforts are appreciated. I have no further questions.
Reviewer 4 Report
All my concerns have been addressed.